# Rumen Fermentation, Digestive Enzyme Activity, and Bacteria Composition between Pre-Weaning and Post-Weaning Dairy Calves

**DOI:** 10.3390/ani11092527

**Published:** 2021-08-28

**Authors:** Yangyi Hao, Chunyan Guo, Yue Gong, Xiaoge Sun, Wei Wang, Yajing Wang, Hongjian Yang, Zhijun Cao, Shengli Li

**Affiliations:** 1State Key Laboratory of Animal Nutrition, Beijing Engineering Technology Research Center of Raw Milk Quality and Safety Control, College of Animal Science and Technology, China Agricultural University, Beijing 100193, China; B20193040338@cau.edu.cn (Y.H.); gongyue@brightdairy.com (Y.G.); xiaogesun@163.com (X.S.); yajingwang@cau.edu.cn (Y.W.); yang_hongjian@cau.edu.cn (H.Y.); caozhijun@cau.edu.cn (Z.C.); 2Jinzhong Vocational and Technical College, Jinzhong 030024, China; jzzygcy@126.com

**Keywords:** calf, weaning, rumen, fermentation, enzyme, bacteria

## Abstract

**Simple Summary:**

Weaning is very important for young ruminants. At this stage, calves’ main source of nutrients is transferred from milk into solid feed, such as starter and roughage. At the same time, the rumen function of calves undergoes tremendous changes, such as bacteria, which are the main players in rumen function. Our research found that the rumen bacteria network of post-weaning calves was more complex. The fermentation end products, such as acetate, propionate, and butyrate, were higher in the post-weaning calves than the pre-weaning group. However, digestive enzymes such as protease, carboxymethyl cellulase, cellobiohydrolase, and glucosidase were lower in the post-weaning calves than the pre-weaning calves. These findings provided useful information for reference regarding the feeding management of calves.

**Abstract:**

To better understand the transition of rumen function during the weaning period in dairy calves, sixteen Holstein dairy calves were selected and divided into two groups: pre-weaning (age = 56 ± 7 day, *n* = 8) and post-weaning (age = 80 ± 6 day, *n* = 8). The rumen fluid was obtained by an oral gastric tube. The rumen fermentation profile, enzyme activity, bacteria composition, and their inter-relationship were investigated. The results indicated that the post-weaning calves had a higher rumen acetate, propionate, butyrate, and microbial crude protein (MCP) than the pre-weaning calves (*p* < 0.05). The rumen pH in the post-weaning calves was lower than the pre-weaning calves (*p* < 0.05). The protease, carboxymethyl cellulase, cellobiohydrolase, and glucosidase in the post-weaning calves had a lower trend than the pre-weaning calves (0.05 < *p* < 0.1). There was no difference in α and β diversity between the two groups. Linear discriminant analysis showed that the phylum of *Fibrobacteres* in the post-weaning group was higher than the pre-weaning group. At the genus level, *Shuttleworthia*, *Rikenellaceae*, *Fibrobacter*, and *Syntrophococcus* could be worked as the unique bacteria in the post-weaning group. The rumen bacteria network node degree in the post-weaning group was higher than the pre-weaning group (16.54 vs. 9.5). The *Shuttleworthia* genus was highly positively correlated with MCP, propionate, total volatile fatty acid, glucosidase, acetate, and butyrate (r > 0.65, and *p* < 0.01). Our study provided new information about the rumen enzyme activity and its relationship with bacteria, which help us to better understand the effects of weaning on the rumen function.

## 1. Introduction

Weaning, the most stressful and significant transition experienced by dairy calves, influences the ability of calves to adapt to the dramatic dietary shift and the production performance. The rumen becomes much more susceptible to changes in fermentability of diets after weaning [1]. The development of ruminal bacteria is also affected by weaning, and even a perfect weaning strategy cannot cover the interference of this transition [2]. McCurdy et al. demonstrated that the ability of the rumen to manage rumen pH changes fundamentally post-weaning [1]. Rumen buffering ability and metabolic adaptations increased after weaning, along with an increase in feed intake, especially the roughage [3,4,5]. Chong et al. indicated that weaning could change the rumen fermentation profile and alter the rumen microbiota composition; the functional development of the rumen is influenced by weaning, which is also associated with ruminal microbiota in lambs [6]. Oral inoculation of pre-weaned young calves with rumen microbiota from adult cows could affect the colonization of some rumen bacteria, methanogens, and protozoa, and some metabolic pathways, which showed that the regulation of rumen microbiota may be a promising way to shape the calves’ rumen function [7]. In reality, rumen microbiota and fermentation progress are a comprehensive category, which needs further in-depth and detailed research.

Bacteria is the most abundant microorganism and the major contributor to digesting plants in the rumen, accounting for about 95% of the microbiota [8,9]. Enzymes, which are encoded and secreted by microorganisms, play a primary role in plant degradation [10,11]. Multiple glycoside hydrolases work to deconstruct the intricate chemical structure of plant biomass; these enzymatic functions are in fact translated into microbes playing a role within the rumen ecosystem [12]. Lipases can regulate fatty acid metabolism in the rumen, and control of lipolysis in the rumen could play a vital role in limiting the biohydrogenation of polyunsaturated fatty acids [13]. Xylanase breaks down the polysaccharide β-1,4-xylan, a main chain of hemicellulose, while carboxymethyl cellulase (CMCase) attacks low-crystalline regions in the cellulose fiber, creating free chain ends by degrading β-1,4-glucan [14]. Dehydrogenase, urease, and protease can react with protein and urea, which further supplies ideal protein nutrients to the host [15,16,17]. Meanwhile, the rumen microbiota plays a crucial role in digesting the feed into volatile fatty acids (VFA), ammonia, and microbial crude protein (MCP), which can further supply nutrients for ruminants [18,19]. In the past few decades, many studies have focused on the effects of nutrients, such as starter composition, minerals, vitamins, etc., on the weaning calves’ productive performance, and these issues have been explained clearly [20]. Recently, studies indicated that the rumen bacteria community had an oscillation during the weaning period, which further influenced the establishment of a stable rumen bacteria community [21,22]. However, few studies have paid attention to the variation in the activity of rumen digestive enzymes during the weaning period. There is still a shortage of a clear understanding of the variation and relationships within the rumen fermentation profile, digestive enzyme activity, and bacteria community during the weaning period, which inhibit the precise feeding of weaned calves.

The objective of this study was to investigate the ruminal fermentation profile, digestive enzyme activity, and bacteria composition in pre- and post-weaning calves; the bacteria network and the relationship among the rumen bacteria, enzyme activities, and fermentation indexes have been analyzed. We hope to illustrate the variation in rumen function during the weaning transition period, further clarifying the dominant functions of the weaning calves’ rumen. Ultimately, we aim to provide a reasonable basis for calf feeding and management.

## 2. Materials and Methods

### 2.1. Experiment Design and Animal Management

A total of sixteen female Holstein dairy calves were selected, with a standard of healthy, half-sibs, and no antibiotic use a month before sampling, and were divided into two groups: pre-weaning (age = 56 ± 7 day, *n* = 8) and post-weaning (age = 80 ± 6 day, *n* = 8). The calves were weaned at the age of 60 days, and the post-weaning group was about twenty days after weaning. The pre-weaning group was fed 5 L of milk by artificial feeding each day, and both of the groups had free access to a mix of starter and oat grass (the starter-to-oat-grass ratio was 85:15, and the oat was pre-chopped into a length of 4–5 cm), and water. The ingredients of the starter were composed of corn grain, soybean, minerals, and vitamins, which qualified for the calves’ nutrient requirements; detailed information about the diet is shown in Appendix A. Each animal was separated from her mother after birth and fed in a separate hutch for about 90 days. The experimental procedures were approved by the Ethical Committee of the College of Animal Science and Technology, China Agricultural University (protocol number: 2013-5-LZ). 

### 2.2. Sample Collection and Measurement

#### 2.2.1. Rumen Fluid Collection and Measurement

Rumen fluid was collected only once by an oral gastric tube (Ancitech, Winnipeg, MB, Canada) one hour before morning feeding. The initial 50 mL of rumen fluid was discarded to avoid saliva contamination, and finally, 50 mL of rumen fluid was filtered by four layers of cheesecloth and then collected. The rumen fluid pH was immediately measured using a pH electrode (model pH B-4; Shanghai Chemical, Shanghai, China) after collection. The rumen fluid sample of each calf was separated into two parts: one was directly placed in liquid nitrogen and then transferred to −80 °C for 16S rRNA sequencing and enzyme activity analysis; the other was stored at −20 °C for VFA, MCP, and NH_3_-N analysis.

An amount of 0.2 mL of 25% metaphosphoric acid was added to 1.0 mL of the rumen fluid samples (to wipe off the albumen precipitation) before quantification of VFAs and NH_3_-N. The NH_3_-N concentration was measured using the phenol-sodium hypochlorite colorimetry method described by Broderick and Kang [23]. The MCP concentration was detected by referring to Negi et al. [24]. The VFA content was measured using gas chromatography (6890 N; Agilent technologies, Avondale, PA, USA) by referring to Cao et al. [25]. The urease [17], CMCase [26], protease [27], cellobiohydrolase [28], amylase [29], glucosidase [30], lipase [13], xylanase [31], and dehydrogenase [32] were measured using the SpectraMax 190 Microplate Reader (MD., New York, NY, USA) and commercial kits (Suzhou Grace Biotechnology Co., Ltd., Jiangsu, China). Specifically, the rumen fluid was centrifuged at 140× *g* for 10 min at 4 °C, and the supernatant was ultrasonically broken for 3 min. It was then centrifuged at 13,780× *g* for 5 min, and finally, the supernatant fluid was used for measurement. The principle of enzyme activity determination is to carry out an enzymatic reaction under a particular substrate condition and then determine the concentration of the specified end product. The catalytic substrate and reaction process corresponding to each enzyme are described in Appendix A.

#### 2.2.2. 16S rRNA Sequencing

The DNA of rumen fluid samples was extracted using HiPure Stool DNA Kits (Magen company, Guang Zhou, China). The quality of DNA was confirmed by 1% agarose gel electrophoresis. The V3-V4 region of the 16S rRNA gene was amplified by PCR (denaturation: 94 °C for 2 min, followed by 30 cycles at 98 °C for 10 s, annealing reaction: 62 °C for 30 s and 68 °C for 30 s, and a final extension at 68 °C for 5 min) using the former primer 341F (CCTACGGGNGGCWGCAG) and the reverse primer 806R (GGACTACHVGGGTATCTAAT) [33]. Amplicons were extracted from 2% agarose gels and purified using the AxyPrep DNA Gel Extraction Kit (Axygen Biosciences, Union City, CA, USA). The amplicons were quantified using an ABI StepOnePlus Real-Time PCR System (Life Technologies, Foster City, CA, USA). All of the purified amplicons were pooled in equimolar amounts and paired-end sequencing on a PE250 Illumina platform, and each sample was identified by its special barcode. Paired-end reads were merged using FLASH (V1.2.7, http://ccb.jhu.edu/software/FLASH/, accessed date 1 October 2017) [34]. Low quality (score ≤ 20), short reads (<200 bp), and reads containing ambiguous bases or unmatched to primer sequences and barcode tags were filtered to obtain the high-quality clean tags [35] according to the QIIME (V1.9.1, http://qiime.org/scripts/split_libraries_fastq.html, accessed date 14 May 2018) [36] quality-controlled process. The tags were compared with the reference database (Silva database, https://www.arb-silva.de/, accessed date 1 September 2019) using the UCHIME algorithm (UCHIME Algorithm, http://www.drive5.com/usearch/manual/uchime_algo.html, accessed date 1 October 2017) [37] to remove the chimera sequences [38], and the effective tags were finally obtained. Sequence analysis was performed by Uparse software (Uparse v7.0.1001, http://drive5.com/uparse/, accessed date 1 October 2017) [39]. Sequences with ≥97% similarity were assigned to the same Operational Taxonomic Units (OTUs). The representative sequence for each OTU was screened for further annotation. OTU abundance information was normalized using a standard sequence number corresponding to the sample with the least sequences. For each representative sequence, the Silva Database 132 (http://www.arb-silva.de/, accessed date 1 September 2019) [40] was used based on the mothur algorithm to annotate taxonomic information. Subsequent analysis of alpha- and beta diversity was performed based on this normalized data output.

### 2.3. Statistics

The rumen fermentation profile and enzyme activity were analyzed using a one-way ANOVA followed by a *t*-test in SAS (SAS version 9.4, SAS Institute Inc., Cary, NC, USA). Contrasts were considered significant when the *p*-value was ≤0.05, and as the trend when the *p*-value was >0.05 and ≤0.10.

Alpha-diversity indices were calculated with QIIME (Version 1.7.0) and analyzed using the Wilcoxon rank-sum test using the “dplyr” package in R (Version 3.6.1). Principal components analysis (PCA) and analysis of similarities (ANOSIM) (999 permutations) were performed and visualized using the “factoextra” package in R (Version 3.6.1). The linear discriminant analysis effect size (LEfSe) was used to determine the difference in rumen bacteria between the two groups. Spearman’s rank correlation was used to identify the relationship among the different genera, the genera with enzymes, VFA, MCP, NH_3_-N, and pH, using the “corrplot” package in R. The result was visualized as a network figure using Gephi (version 0.9.1, Institute Inc., Paris, French) and a heatmap using the R package “heatmap.” All *p*-values were corrected using a false discovery rate of 0.05, as described by Benjamini and Hochberg [41]. The false discovery rate corrected *p* < 0.05 was considered significant.

## 3. Results

### 3.1. Fermentation Profile and Digestive Enzyme Activity

The rumen pH and acetate-to-propionate ratio (A/P) in the post-weaning calves were lower (*p* < 0.01) than the pre-weaning calves (Table 1). However, MCP, acetate, propionate, butyrate, and TVFA in the post-weaning calves were higher (*p* ≤ 0.012) than the pre-weaning calves, with no difference (*p* ≥ 0.38) in the NH_3_-N, valerate, and isovalerate concentrations.

In the post-weaning group, a trend of lower concentrations was detected for protease (26.9%, *p* = 0.063), CMCase (33.1%, *p* = 0.085), and cellobiohydrolase (33.7%, *p* = 0.088), but a greater concentration (39.6%, *p* = 0.065) for glucosidase (Table 2).

### 3.2. Rumen Bacteria Composition

All of the samples with a coverage rate >99.9% and the α diversity indexes including Ace, Chao1, Shannon, and Simpson had no difference between the two groups (Table 3). The Venn diagram showed that 179 genera were shared between the two groups, and the pre- and post-weaning groups had 55 and 75 unique genera, respectively (Figure 1a). The principal coordinate analysis showed parts of the two groups separated from each other, with 25.77% and 20.31% of variations explained by the principal components PC1 and PC2, respectively. ANOSIM showed that the two groups had no statistical difference (R^2^ = 0.04, *p* = 0.21).

The top-ten phyla account for more than 99% of the bacteria, and the *Bacteroidota*, *Firmicutes*, and *Proteobacteria* were the top-three phyla (Figure 2a). The top-fifteen genera are shown in Figure 2b, and the *Prevotella* was the most abundant genus.

A total of 22 bacteria clusters were identified as biomarkers of the two groups by the Lefse analysis with a linear discriminant analysis score of >2.5 (Figure 3). The phylum *Fibrobacteres* was higher in the post-weaning group than the pre-weaning group; however, the *Synergistates* and *Melainabacteria* phyla were lower than in the pre-weaning group. At the genus level, *Pyramidobacter* was higher in the pre-weaning group than the post-weaning group; the *Rikenellaceae*, *Fibrobacter*, *Syntrophococcus*, and *Shuttleworthia* genera could be worked as the unique bacteria in the post-weaning group.

### 3.3. The Network and Correlation between Bacteria and Its Byproducts

The connection network showed that the post-weaning group had more rumen bacteria complexity than the pre-weaning group, which is indicated by the higher node degree in the post-weaning group than the pre-weaning group (16.54 vs. 9.5) (Figure 4). The node represents the correlation of a genus with other genera (r > 0.6 or < −0.6 and *p*-value < 0.05). The node and edge numbers in the pre- and post-weaning groups were 234 and 254, and 2223 and 4200, respectively. The positive correlation (r > 0.6 and *p* < 0.06) percentage in the post-weaning group (94.54%) was higher than the pre-weaning group (82.55%). There was one clustered genus in the pre-weaning group; however, the post-weaning group had two clustered genera, and the number of other genera was highly correlated in the center of the network.

The correlation between the top-30 most-abundant bacteria genera and enzymes, VFA, MCP, NH_3_-N, and pH is shown in Figure 5. A total of 31 positive correlations (*p* < 0.05) and 36 negative correlations (*p* < 0.05) were found. The *Shuttleworthia* genus was highly positively correlated with MCP, propionate, TVFA, glucosidase, acetate, and butyrate (r > 0.65, and *p* < 0.01). The dehydrogenase was highly positive correlated with *Lachnospiraceae_NK3A20_group* and *Acetitomaculum* (r > 0.68 and *p* < 0.01). The rumen pH was highly negatively correlated with *Syntrophococcus*, *Shuttleworthia*, and *Erysipelotrichaceae_UCG-009* (r < −0.68 and *p* < 0.01).

## 4. Discussion

### 4.1. Rumen Fermentation Profile and Enzyme Activity

VFAs are the end products of diet fermentation, and they are also essential for rumen development, production performance, and body metabolism [42,43,44]. Our results, which showed that acetate, propionate, and butyrate were higher in the post-weaning calves than the pre-weaning calves, and indicated that the post-weaning calves’ rumen could produce more VFA for the host to utilize, were consistent with Kong et al. [45]. Of course, the rumen VFA content is also highly correlated with feed intake; as such, the lack of feed intake data to verify this point is a defect of our study. We deduced that the higher feed intake of the post-weaning calves was also contributed to by the rumen VFA content. The rumen pH was influenced not only by differences in the feed, saliva, and rate of passage in the rumen, but also by fermentation products, such as propionate, acetate, and NH_3_-N [46]. The higher TVFA in the post-weaning calves made their rumen pH lower than the pre-weaning calves. Non-protein nitrogen could be hydrolyzed into ammonia by urease produced by microbes [17]. The protein is hydrolyzed into amino acids and peptides by protease, and then parts of amino acids also become ammonia by microbial deaminating [17]. A portion of ammonia synthesis produces MCP via microorganisms, the other parts being absorbed into the blood, participating in the rumen nitrogen cycle [47]. MCP is an ideal protein source for ruminants [47]. NH_3_-N showed no difference between the two groups, but the MCP in the post-weaning group was higher than the pre-weaning group, which may be due to the microbiota in the post-weaning group having a higher ability to synthesize MCP.

Xylanase and cellulose are the most popular exogenous fibrolytic enzymes used in ruminants [48]. The exogenous fibrolytic enzymes could improve the NDF digestibility, indicating that under certain circumstances (i.e., in a poorer rumen development, ruminants with high-grain diets) the fibrinolytic enzyme is limited in the rumen [49,50]. Pre-weaning calves tended to show a greater concentration of CMCase, cellobiohydrolase, and glucosidase (which belong to the exogenous fibrolytic enzymes) than the post-weaning calves, which also indicated that the rumen fiber decomposition ability in the post-weaning calves was damaged. The protein is hydrolyzed into amino acids and peptides by protease, and then parts of amino acids also became ammonia by microbial deaminating [17]. However, protease tended to decrease in post-weaning calves, which may be a signal that we need to pay special attention to the protein quality and digestibility in post-weaning calves’ diet. The stress of weaning and relying on rumen to provide nutrients to the host is a challenge for post-weaning calves. The post-weaning calves need more attention and improving the feed quality to ensure post-weaning calves have enough nutrients to maintain growth and health is vital.

### 4.2. Rumen Bacteria Composition and Its Effects on Rumen Function

Under the same diet conditions, the rumen microbiota composition and function were the difference between the high and low feed-efficiency cattle [51,52]. The pre- and post-weaning calves’ rumen bacteria α and β diversity showed no difference. However, the Venn analysis results indicated that there were still some differences between the two groups; the distinct bacteria may shape the different rumen functions. *Fibrobacter*, which can encode carbohydrate-active enzyme families involved in the plant cell wall polysaccharide degradation, has an important role in rumen fiber digestion [53]. Acetate is the end product of fiber decomposition, and a high fiber content diet could also enhance the rumen acetate concentration [54]. It was consistent that the fiber degradation-related bacteria, such as *Fibrobacter*, and acetate were higher in the post-weaning calves than the pre-weaning calves. The bacteria may be the reason for the high rumen acetate content. The *Shuttleworthia* genus was abundant in the high hay diet, which was highly correlated with fiber digestion [55]. *Shuttleworthia* was the unique bacteria in the post-weaning group and highly correlated with several key rumen digestive enzymes and fermentation indexes, which make it a promising target bacteria to regulate rumen function. However, the discrepancy between the fibrolytic enzymes and the VFA needs to be further explored. Here, we deduced that it may be due to the post-weaning calves having a higher feed intake, which also contributed to the VFA production. Meanwhile, there may be an imbalance between VFA production and rumen epithelium absorption, which leads to VFA accumulation in post-weaning calves rumen [56]. The lower rumen pH in the post-weaning groups also verified this point.

Quorum sensing is a well-recognized form of bacterial communication, rife in the rumen bacteria [57]. In fact, one bacteria could have multiple functions; the rumen digestion was the result of microbiota synergy. Rumen bacteria could be divided into a series of functional groups [58]. The post-weaning calves had a more complicated rumen bacteria network, and it seems that the bacteria clustered into several different parts; this may be due to the weaning disturbing the rumen bacteria network. There was a study that demonstrated the simple bacteria complexing and it was argued that low richness bacteria could improve rumen energy harvestability [59]. However, not all microbiota favor the rumen energy harvest, as there is a lot of functional redundancy in the rumen ecosystem [60]. The core rumen microbiota dictates ruminant feed efficiency and productivity [61]. The disturbance of the rumen bacteria community’s normal order may lead the post-weaning calves’ rumen bacteria network to be more complicated and down-regulate some digestive enzyme activity.

## 5. Conclusions

Weaning has an extensive influence on rumen function. Our results indicated that the post-weaning calves had a stronger rumen fermentation ability, which was verified by the higher acetate, propionate, and butyrate content. However, the enzyme activity of protease, CMCase, cellobiohydrolase, and glucosidase in the post-weaning calves had a lower trend than in the pre-weaning calves. The post-weaning calves have a more complicated rumen bacteria network than pre-weaning calves. *Fibrobacter* and *Shuttleworthia* were higher in the post-weaning calves than the pre-weaning calves. *Shuttleworthia* was also highly correlated with several key digestive enzymes and VFAs, which could make it a promising target bacteria for regulating weaned calves’ rumen function. Our study has provided a piece of information to better understand the rumen function of weaning calves.

## Figures and Tables

**Figure 1 animals-11-02527-f001:**
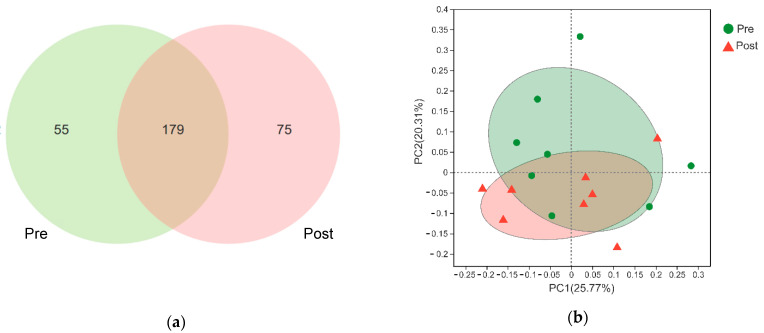
(**a**) Venn analysis of the shared and unique bacteria at the genus level. (**b**) Principal components analysis of the rumen bacteria community among the two groups. Pre: pre-weaning group; Post: post-weaning group.

**Figure 2 animals-11-02527-f002:**
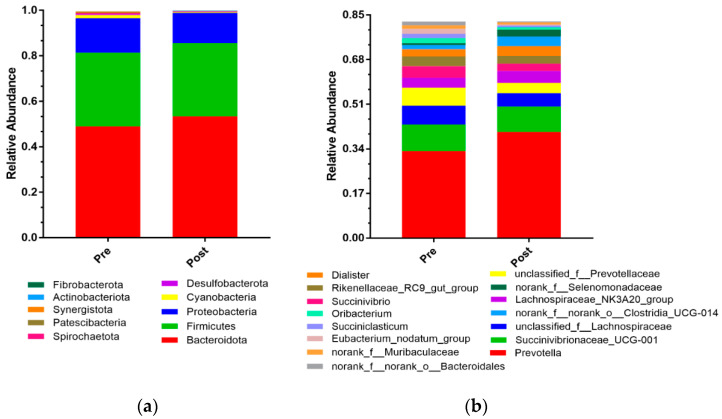
Histograms of the relative abundances of rumen microbial taxa at different ages. (**a**) At the phylum level (top ten). (**b**) At the genus level (top fifteen).

**Figure 3 animals-11-02527-f003:**
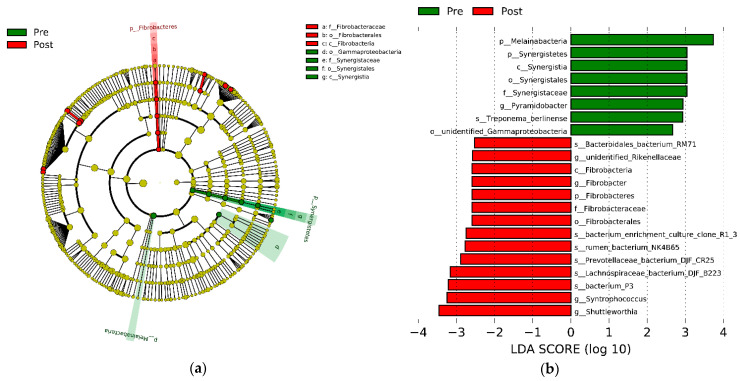
The linear discriminant analysis effect size (LEfSe) analysis of bacterial taxa within the two groups. (**a**) Cladogram shows significantly enriched bacterial taxa (from the phylum to genus level). (**b**) Histogram of the linear discriminant analysis (LDA) scores computed for differentially abundant rumen bacteria between the pre- and post-weaning groups. Significant differences are defined as *p* < 0.05 and an LDA of score >2.5. Pre: pre-weaning group; Post: post-weaning group.

**Figure 4 animals-11-02527-f004:**
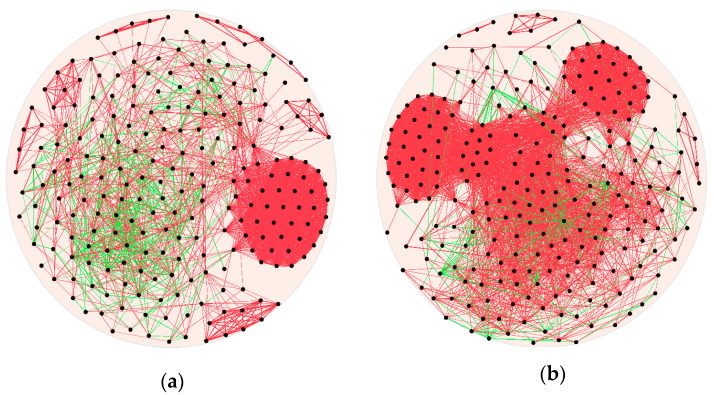
Bacteria community network analysis was used to explore the relationship between different genera. (**a**) Pre-weaning group. (**b**) Post-weaning group. The node represented the genus, which has a Spearman rank correlation coefficient of >0.6 or <−0.6. The thickness of the edges represents the correlation coefficient; the thicker the edge, the greater the correlation (all of the correlations were r > 0.6 or < −0.6, *p* < 0.05). The green line indicates a negative correlation, and the red line shows a positive correlation.

**Figure 5 animals-11-02527-f005:**
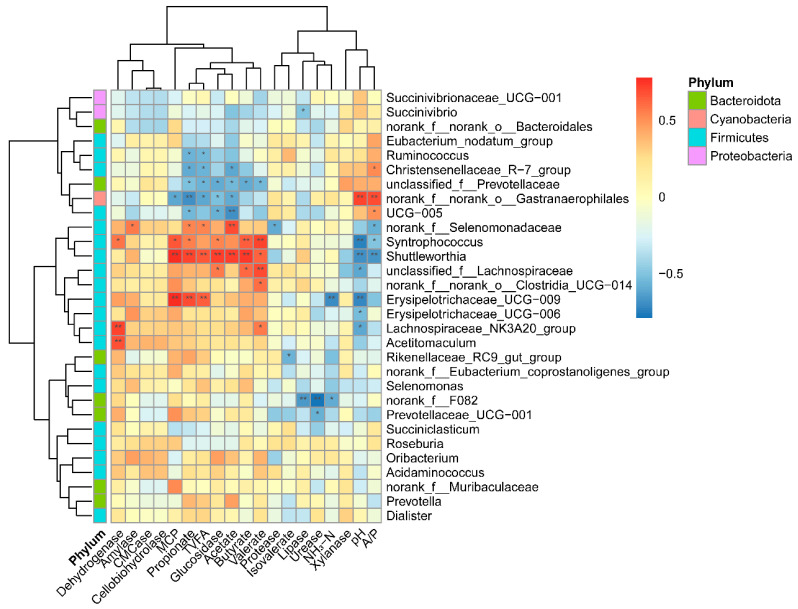
The correlation between bacteria (genus level) and enzymes, VFA, MCP, NH_3_-N, and pH. Cells are colored based on Spearman’s correlation coefficient: red represents a positive correlation, and blue represents a negative correlation. The darker the color, the stronger the correlation. “*” and “**” indicate false discovery rate (FDR)-adjusted *p*-values < 0.05, and <0.01, respectively. NH_3_-N: ammonium nitrogen; MCP: microbial crude protein; VFA: volatile fatty acid; TVFA: total volatile fatty acid; A/P: the ratio of acetate to propionate.

**Table 1 animals-11-02527-t001:** The effects of weaning on the rumen fermentation profile of Holstein dairy calves.

Items ^1^	Pre	Post	SEM	*p*
pH	6.63	6.18	0.08	0.003
NH_3_-N (mg/dL)	7.68	8.78	1.94	0.711
MCP (μg/mL)	109.43	157.72	6.41	<0.001
Acetate (mmol/L)	44.73	59.62	2.95	0.012
Propionate (mmol/L)	24.26	50.94	4.26	0.001
Butyrate (mmol/L)	4.21	9.19	0.87	0.001
Isovalerate (mmol/L)	0.73	0.58	0.07	0.650
Valerate (mmol/L)	1.06	2.25	0.26	0.383
TVFA (mmol/L)	80.61	124.22	6.96	0.001
A/P	2.11	1.26	0.13	<0.001

^1^ NH_3_-N: ammonium nitrogen; MCP: microbial crude protein; VFA: volatile fatty acid; TVFA: total volatile fatty acid; A/P: the ratio of acetate to propionate; SEM: standard error of the mean; Pre: pre-weaning group; Post: post-weaning group.

**Table 2 animals-11-02527-t002:** The effects of weaning on the digestive enzyme activity of Holstein dairy calves.

Items ^1^	Pre	Post	SEM	*p*
Dehydrogenase (μg/min/mL)	0.76	0.85	0.03	0.139
Urease (μg/min/mL)	3.37	2.86	0.31	0.449
Protease (μg/min/mL)	17.93	13.11	1.27	0.063
CMCase (μg/min/mL)	83.61	55.93	7.78	0.085
Cellobiohydrolase (μg/min/mL)	56.82	37.68	5.43	0.088
Glucosidase (nmol/min/mL)	120.84	200.21	21.00	0.065
Amylase (mg/min/mL)	1.04	1.11	0.13	0.796
Lipase (nmol/min/mL)	160.28	160.10	12.30	0.994
Xylanase (nmol/min/mL)	216.51	200.73	19.24	0.706

^1^ CMCase: carboxymethyl cellulase; SEM: standard error of the mean; Pre: pre-weaning group; Post: post-weaning group.

**Table 3 animals-11-02527-t003:** The effects of weaning on the rumen bacteria α-diversity of Holstein dairy calves.

Items ^1^	Pre	Post	SEM	*p*
Sobs	124.38	131.13	6.89	0.75
Ace	127.42	134.08	6.84	0.79
Chao1	126.53	133.70	6.83	0.87
Shannon	2.52	2.52	0.10	0.96
Simpson	0.18	0.21	0.02	0.79

^1^ Pre: pre-weaning group; Post: post-weaning group; SEM: standard error of the mean.

## Data Availability

The datasets analyzed are not publicly available due to ownership by the funding partners, but are available from the corresponding author on reasonable request.

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
