# Peer review of "Rumen Fermentation, Digestive Enzyme Activity, and Bacteria Composition between Pre-Weaning and Post-Weaning Dairy Calves"

_animals, 2021, doi:10.3390/ani11092527_

Round 1

Reviewer 1 Report

Dear authors,

The overall idea is of high interest and I would like to see such work be published. The authors made an interesting conceptualization that was combined with a strong set of methods and could provide plenty of novel results. The topic is very trend since the manipulation of rumen biochemistry in early life appears to be the stepping stone for adults' productivity. However, the results were not presented with consistency and the discussion did not strongly support the study findings giving the sense of incomplete work. 

Major comments:

Introduction: The introduction should be extensively revised since there is very general, somehow irrelevant information. There is no literature review of previous works on the topic and the novelty of the present study. What has been done up to date and what your study comes to bridge? This must be included.

Material and methods: Material and methods are very weak. Many information should be added. More specifically:

1) Did you add any acidifier in samples for VFA and NH3-N?
2) Did you use cheesecloth for rumen fluid? how many layers? 
3) Lines 90-97: Working on rumen enzyme activity as well, there is a serious lack of literature about well-documented assays for enzymes activities in the rumen. This part of your study is the most significant for many researchers. These protocols should be extensively described providing all the details no matter how long the section will be. 
4) Line 108: Did you pool the DNA samples of each group?
5) Table for feed composition is missing. 
6) Data about experimental duration and sampling procedure are missing too (number of animals and sampling times).
7) Did calves suck their mothers (at any age)? It is quite significant data since there is a horizontal transfer of the rumen microbiome. 

Major concern: The presence of a starter in pre-weaning makes me wonder about the precision of the experimental design. Since animals' individuality could affect feed consumption and consequently the rumen microbiome, the results may have been affected as well. I would like to hear your opinion.

Results: Lines 162-166. Seems more materials than results. 
Figure 4. Very nice approach. Please focus more on results like this. 
Line 276: top 30 bacteria: please clarify; top 30 based on their abundance?

Discussion: 
Protease tended to decrease. Make some assumptions about this trend.
Lines: 312-319: Wrong results about cellulase. CMCase tended to decrease. Please check again your results and discuss them accordingly. 

Lines: 331-335. Due to CMCase activity, these statements should be re-evaluated.

Author Response

 Dear reviewer1:
Thanks for your comments on our manuscript and please see the attachment for the reply to your comments.

Thank you again

 Best wishes

Yangyi Hao

Reviewer 2 Report

The purpose of this study was to compare some variables of rumen fermentation, digestive enzyme activity, and bacteria composition between pre-weaning and post-weaning dairy calves. The subject falls within the specific scope of special issue "rumen function". The manuscript reports a topic pertinent to contemporary. The manuscript is well written and organized. The statistical analyses are adequate and report a quality data. There are few little flaws which should be rectified before publication.

Title: is OK, but I think could be more specific, for example: “Rumen fermentation, digestive enzyme activity, and bacteria composition between pre-weaning and post-weaning dairy calves”

Others observations

L75: Please, describe better the background of experimental units. For example. How many days, the post-weaning group were only fed with starter plus grass ((were withdrew from the liquid diet) before the sample was taken. What was the intake level of starter and forage for each group at the moment of sampling? Forage was offered chopped?

L76: Please, indicate (at least) CP level and NDF content of calf-starter and oat grass.

L78: All animals were maintained in individual cages up to they were sampled? Or only pre weaning group was maintained in individual cages?

L83: Please clarify if the ruminal fluid sample was taken only once, or the sample was taken on consecutive days and the final sample represents a composite sample of several days.

L125: Given the low number of observations per treatment, all data should have passed normality first.

L127: Please rewording as: Contrasts are considered significant when the P value was ≤ 0.05, and as trend when the P-value was > 0.05 and ≤ 0.10.

Table 1 and Table 2: Since the p-value is indicated in right column of Tables, the footnote legend "The different lowercase letter between the same columns means that the difference is significant (P < 0.05). 0.05 < P < 0.1  indicated the two groups had a different trend" it hurts.  Please, remove.

Please, put P-value after comparison For example: L148: were lower (P<0.001) than pre-weaning calves

L149: Please rewording as:  propionate, butyrate, and TVFA in the post-weaning calves were higher (P≤0.012) than pre-weaning calves, without difference (P≥0.38) on NH3-N, valerate and isovalerate concentrations.

L157: Please rewording as: In post-weaning group, a trend for lower concentration for protease (26.9%, P=0.063), CMCase (33.1%, P=0.085), and cellobiohydrolase (33.7%, P=0.088) were detected, but a greater concentration (39.6%, P=0.065) for glucosidase.

L294: Please, mention here that VFA influencing the proliferation and differentiation of epithelial rumen cells as well

L312-314: This generalized statement must be specified as: The exogenous fibrolytic enzymes could improve the neutral detergent fiber digestibility indicate that, under certain circumstances (i.e. in a poorer rumen development, ruminants with high-grain diets) the fibrinolytic enzyme is limited in the rumen.

L316: Please rewording as: Post-weaning calves tended to show greater concentration of CMCase, cellobiohydrolase, and glucosidase (which belong to the exogenous fibrolytic enzymes) than the pre-weaning calves, which also indicated the rumen feed decomposition ability in the weaned calves was more robust.

L321-322: Please rewording this statement (is confuse)

Author Response

Dear reviewer2:
Thanks for your comments on our manuscript and please see the attachment for the reply to your comments.

Thank you again

 Best wishes

Yangyi Hao

Round 2

Reviewer 1 Report

Dear authors,
The manuscript has been substantially improved but there are still comments that have not been completely addressed (enzymes assays). Please perform the appropriate correction. 

Point 1: Introduction: The introduction should be extensively revised since there is very general, somehow irrelevant information. There is no literature review of previous works on the topic and the novelty of the present study. What has been done up to date and what your study comes to bridge? This must be included.

Reply: Dear reviewer, thanks for your comments on our introduction parts and we have reorganized this part and added some reviews, which you have pointed are necessary for this part.

The first paragraph of the introduction was to introduce the characters of the calves’ rumen and highlight the importance of weaning on the calves.

In the second paragraph, we introduced the role of bacteria, enzymes, and VFAs roles for rumen function and host. And then we follow your advice to introduce some reviews of previous works on the topic, the novelty of the present study, and our study comes to a bridge. The second paragraph contents were as follows: “Bacteria is the most abundant microorganism and the major contributor to digesting plants in the rumen, account for about 95% of the microbiota [3,4]. The primary role for plant degradation is the enzymes, which were encoded and secreted by the microorganisms [5,6]. Meanwhile, the rumen microbiota plays a crucial role in digest the feed into volatile fatty acids (VFA), ammonia, and microbial crude protein (MCP), which

R1: The introduction was not extensively revised or re-organized. However, the addition is satisfactory.

Point 2: Material and methods: Material and methods are very weak. Many information should be added. More specifically:

Did you add any acidifier in samples for VFA and NH3-N?

Reply: Yes, we added the acidifier. 

R1: Thank you.

Did you use cheesecloth for rumen fluid? how many layers?

Reply: we use the cheesecloth for the rumen fluid collection and it was four layers. The specific information was added to the manuscript as follows: “…50 mL rumen fluid was filtered

R1: Thank you.

Lines 90-97: Working on rumen enzyme activity as well, there is a serious lack of literature about well-documented assays for enzymes activities in the rumen. This part of your study is the most significant for many researchers. These protocols should be extensively described providing all the details no matter how long the section will

Reply: Dear Reviewer, Thanks for your advice and we have revised this part as your suggestion. The revised version was as follows: “The urease [15], carboxymethyl cellulase (CMCase) [16], protease [17], cellobiohydrolase [18], amylase [19], glucosidase [20], lipase [21], xylanase [21], and dehydrogenase [22] were measured using the SpectraMax 190 Microplate Reader (MD., Newyork, USA) with the commercial kits (Suzhou Grace Biotechnology Co., Ltd, Jiangsu, China),........

R1: I greatly regret to say that the current version is not satisfactory. The borderline which discriminates the researchers/scientists from the technicians is the understanding of methods’ principles. Many of your references in enzymes are extremely old-fashioned and I doubt about their reliability (especially for the lipase which appears to be a questionable assay). I am suggesting at least to describe the principle of each method and provide the reference number of each commercial kit that was used. 

Line 108: Did you pool the DNA samples of each group?

Reply: we pooled all the DNA samples together. The high-throughput sequence technology was using the special biomarker to mark each sample and all the samples pooled together to take sequencing. The manuscript was described as follows. “ All the purified amplicons were pooled in equimolar and paired-end sequenced on a PE250 Illumina platform, each sample was.....

R1: Thank you for the clarification.

Data about experimental duration and sampling procedure are missing too (number of animals and sampling times).

Reply: A total of sixteen calves were included in the experiment and the rumen fluid was sampled once in each group. But the animals were under monitoring for a month to make sure there was no antibiotic be used before the sampling. The specific information was as follows in the manuscript.

R1: Thank you for the clarification.

Did calves suck their mothers (at any age)? It is quite significant data since there is a horizontal transfer of the rumen microbiome.

Reply: We agree with you that there is a horizontal transfer of the rumen microbiome by adult animals. In this study, the calves were separated from their mother after birth and were fed milk by artificial. We revised this part in the manuscript as follows:....

R1: Thank you for the clarification.

Point 3: Major concern: The presence of a starter in pre-weaning makes me wonder about the precision of the experimental design. Since animals' individuality could affect feed consumption and consequently the rumen microbiome, the results may have been affected as well. I would like to hear your opinion.

Reply: dear reviewer, we think it’s a good question for us. The feed intake could affect the rumen microbiome. If we did not provide the calves starter before weaning and gave the starter after weaning, it was obvious the rumen environment and microbiome will have great variation. But under this condition, there may have two treatments, weaning, and starter feeding.

Besides, ruminants have four stomachs, esophageal sulcus reflex could transfer the milk directedly into the abomasum. The rumen is the main feed digestion stomach after weaning. If calves did not have the starter before weaning, it would be difficult for the calves to adapt to the weaning diet. The undeveloped rumen can't ferment, digest, and absorb the nutrients for calves. And the small volume of rumen may not capable to accommodate the feedstuffs.

Therefore, the dairy calves will be provided starter and oat grass before weaning to train their feed intake skill and make the rumen prepare for the weaning diet.

R1: I would agree.

Point 4: Results: Lines 162-166. Seems more materials than results.

Reply: We have revised this part and deleted the materials-related contents and make the results more focused.

R1: Thank you.

Point 6: Line 276: top 30 bacteria: please clarify; top 30 based on their abundance?

Reply: It was the top 30 bacteria in abundance. We have revised the manuscript as follows:” The correlation between the top 30 bacteria genera in abundance and enzyme, VFA, MCP, NH3-N, and pH was shown in Figure 5.”

R1: Thank you for the clarification.

Point 7: Protease tended to decrease. Make some assumptions about this trend.

Lines: 312-319: Wrong results about cellulase. CMCase tended to decrease. Please check again your results and discuss them accordingly.

Reply: Dear reviewer, thanks for your careful review. We have added the discussion about the protease and revised the results about the cellulase. We have revised this parts as follows:” Xylanase and cellulose are the most popular exogenous fibrolytic enzyme used in ruminants [39]. The exogenous fibrolytic enzymes could improve the NDF digestibility indicate that, under certain circumstances (i.e. in a poorer rumen development, ruminants with high-grain diets) the fibrinolytic enzyme is limited in the rumen [40,41]. Pre-weaning calves tended to show a greater concentration of CMCase, cellobiohydrolase, and glucosidase (which belong to the exogenous fibrolytic enzyme) than the post-weaning calves, which also indicated the rumen fiber decomposition ability in the post-weaning calves was damaged. The protein isweaning calves need more attention and improve the feedstuff quality to make sure the post- weaning calves have enough nutrients to maintain growth and health.”

R1: Thank you.

Point 8: Lines: 331-335. Due to CMCase activity, these statements should be re-evaluated.

Reply: Dear reviewer, thanks for your kindly remind and we have revised and added some discussion in this part, the added part was as follows.  “However, the discrepancy between the fibrolytic enzymes and the VFAs needs to be further explored. Here we deduced it may be due to the post-weaning calves had a higher feed intake, which also contributed to the VFAs production. Meanwhile, there may be an imbalance between the VFAs production and rumen epithelium absorption, which led to the VFAs accumulation in post-weaning calves rumen [47]. The lower rumen pH in the post-weaning groups also verified this point.”

R1: Thank you.

The quality of figures has been degraded. Please provide them in high quality.

Author Response

Dear reviewer:

Thank you again for your comments and we have replied to your comments in detail. Please find it in the attachment.

Best wishes
Yangyi Hao

Round 3

Reviewer 1 Report

The article has been improved. I have no more comments.